# Integrative analysis of long non-coding RNA and mRNA in broilers with valgus-varus deformity

**Hehe Tang**[1☯]**, Yaping Guo**[1☯]**, Zhenzhen Zhang**[1]**, Zhuanjian Li**[1,2]**, Yanhua Zhang**[1]**, Yuanfang Li**[1]**, Xiangtao Kang**[1,2]**, Ruili Han**[1,2]*

**1** College of Animal Science and Veterinary Medicine, Henan Agricultural University, Zhengzhou, China,
**2** Henan Innovative Engineering Research Center of Poultry Germplasm Resource, Zhengzhou, China

☯ These authors contributed equally to this work.
* rlhan@126.com

**Data Availability Statement:** All raw data files are available from the NCBI database (project PRJNA644292 with accession number SRR12153415, SRR12153416, SRR12153417, SRR12153418, SRR12153419, SRR12153420.

## Abstract

### Background

Bone abnormality and leg disease in commercial broiler flocks are increasingly prominent, causing serious economic losses to the broiler breeding industry. Valgus-varus deformity (VVD) is a common deformity of the long bone in broilers that manifests as an outward or inward deviation of the tibiotarsus or tarsometatarsus. There is a paucity of studies on the molecular mechanisms of VVD.

### Results

In this study, 6 cDNA libraries were constructed from spleen samples from VVD birds and normal birds. A total of 1951 annotated lncRNAs, 7943 novel lncRNAs and 30252 mRNAs were identified by RNA-sequencing. In addition, 420 differentially expressed (DE) mRNAs and 124 differentially expressed lncRNAs (adjusted *P*-value < 0.05) were obtained. A total of 16 dysregulated genes were confirmed by qPCR to be consistent with the results of the RNA-Seq. The functional lncRNA-mRNA co-expression network was constructed using differentially expressed mRNAs and target genes of the differentially expressed lncRNAs. 11 DE genes were obtained from the analysis. In order to gain insight into the interactions of genes, lncRNAs and pathways associated with VVD, we focused on the following pathways, which are involved in immunity and bone development: the Jak-stat signaling pathway, Toll-like receptor signaling pathway, Wnt-signaling pathway, mTOR signaling pathway, VEGF signaling pathway, Notch signaling pathway, TGF-beta signaling pathway and Fanconi anemia pathway. All together, 30 candidate DE genes were obtained from these pathways. We then analyzed the interaction between the DE genes and their corresponding lncRNAs. From these interaction network analyses we found that *GARS*, *NFIC*, *PIK3R1*, *BMP6*, *NOTCH1*, *ACTB* and *CREBBP* were the key core nodes of these networks.

### Conclusion

This study showed that differentially expressed genes and signaling pathways were related to immunity or bone development. These results increase the understanding of the

**Funding:** This work was supported by the Key Scientific Research Projects of Henan Colleges and Universities (20A230006), the Program for Innovation Research Team of Ministry of Education (IRT16R23), and the Scientific Studio of Zhongyuan Scholars (30601985).

**Competing interests:** The authors declare that the research have no conflicts of interest to disclose.

molecular mechanisms of VVD and provide some reference for the etiology and pathogenesis of VVD.

## Introduction

The global broiler production was reported to be about 100 million tons in 2019, and the main broiler breeds include Arbor Acres broilers, Avian broilers, Ross 308 broilers, Roman broilers, Cobb broilers, Hubbard broilers. Because of its large size, fast growth rate, high uniformity, and high feed conversion characteristics occupy the main market. However, with the development of the breeding industry towards modernization and intensification, the production efficiency of commercial broilers has been greatly improved, and the broiler leg disease has become increasingly prominent [1]. Valgus-varus deformity (VVD) is a common leg deformity in broilers, which is characterized by varus and valgus at a certain angle of the tibiotarsal bone or tarsometatarsus [2]. Valgus-varus angulation was classified as normal, mild deformity, intermediate deformity and severe deformity, (tibia-metatarsus angle deviation less than 5˚ was normal, score = 0; angle between 10˚ and 25˚ was mild deformity, score = 1; angle between 25˚ and 45˚ was moderate deformity, score = 2; angle greater than 45˚ was severe deformity, score = 3) [2, 3]. Affected broilers show abnormalities when standing and walking. Because of lameness or pain, the animal's welfare is seriously affected, as the broiler has difficulty accessing food and water, which leads to weight loss or even death. Inevitably, VVD has caused serious economic losses and crippled the development of global broiler industry [4, 5]. Julian (1984) showed that the incidence of VVD in broilers was 0.5–2%. Cruickshank (1986) showed that the incidence of VVD in broilers was 0.4% at 1 week of age and increased to 5.4% at 5 weeks of age. In our previous study of VVD in Hubbard broilers, the incidence was 1.75% [5–7]. Reduction of the incidence and severity of VVD is an urgent matter. Leterrier and Nys have explored the clinical and anatomical differences of broilers with VVD [8]. We also studied the phenotypes and pathological changes in the early stage of development [7]. The occurrence of VVD may be influenced by many factors, such as the lighting schedule, overfeeding, lack of exercise and genetics [9]. The specific etiology and pathogenesis of VVD has not been clarified. However, previous studies have shown that VVD carries a certain genetic potential, and the incidence of VVD could be reduced by genetic selection [1, 6, 10]. These results suggest that genetic studies will provide a more scientific explanation for the etiology and pathogenesis of VVD.

Long non-coding RNA (lncRNA) is a transcript more than 200 nucleotides in length, without coding potential [11]. lncRNA participates in the regulation of mRNA expression by activating or restraining the ability of protein coding genes [12, 13]. Studies have shown that lncRNA participates in the development of the immune system [14], and also plays important roles in epigenetic regulation, cell cycle regulation, and cell differentiation regulation. In recent years, a growing body of research has uncovered novel lncRNAs and revealed their functions. Several studies have demonstrated that lncRNAs are involved in cellular development and other biological processes, especially in a state of disease [15, 16].

The molecular interaction and regulatory mechanisms in VVD are very limited. The spleen is the largest immune organ of the body, and is the center of cellular immunity and humoral immunity. A number of studies have analyzed the effects of disease on spleen transcriptomics through high-throughput sequencing. As the hub of the neural-endocrine-immune network, the spleen contains a variety of immune active cells and immune factors. Abnormal immune

system function will cause bone metabolism disorder, and then change bone homeostasis, leading to the occurrence of disease. T cells are one of the most important immune cells in the immune response of the spleen, in addition to playing a role in the immune system, it also participates in the process of bone reconstruction, which has a certain influence on the formation of osteoclasts [17, 18]. And previous studies have found that abnormal changes of bone histomorphology in Hubbard VVD broilers. The Hubbard broiler has a high rate of growth and meat production, and the incidence of leg disease is relatively high [7]. In this study, the transcriptome profiles of spleen tissue from 3 normal Hubbard commercial broilers and 3 VVD Hubbard commercial broilers were compared. By analyzing the integrated repertoires and expression patterns of lncRNAs and mRNAs, as well as to identify interactions between lncRNAs and mRNAs through differential expression and co-expression network analysis, candidate target genes associated with VVD were screened out. The results could provide support for exploring the molecular mechanism of VVD and provide reference for healthy leg breeding in broilers.

## Materials and methods

### Animals and management

Hubbard broilers (n = 52,000) were reared at a commercial farm in Tangshan for 40 days. Cage dimensions were 0.8 m long, 0.8 m wide and 0.35 m high. Stocking density was 30 birds when 1 to 9 days old and 12 birds aging 10 to 40 days old. The temperature of the newly hatched chicken house should be 29˚C-33˚C, then it should be reduced 2˚C-3˚C per week, and to 21 ˚C at the age of 4 weeks, and the temperature in the late fattening period should not be lower than 18˚C. In the first two weeks, the relative humidity of the chicken house should be controlled at 65–70%, and in the later period it should be controlled at 55–60%. Lighting was maintained at 60 lux 24L (0 to 5 days), 8 lux 17L (6 to 32 days), and 60 lux 24 L (33 to 40 days). All broilers were fed uniform diets containing 1.0% Ca, 0.45% P, 2,950 kcal/kg ME, and 21.5% CP at age 1 to 18 days; 0.85% Ca, 0.38% P, 3,100 kcal/kg ME, and 20.0% CP at age 19 to 40 days. Broilers were injected with newcastle disease vaccine, mycoplasma vaccine and infectious bursal disease virus vaccine at 1 day old; And immunized with nd-ib vaccine by drinking water at 7 days old and the newcastle disease vaccine was injected at 21 days of age.

### Ethics statement

The experiments were performed according to guidelines and protocols approved by the Institutional Animal Care and Use Committee of Henan Agricultural University, and were approved by the Animal Care Committee of the College of Animal Science and Veterinary Medicine, Henan Agricultural University, China (18–0120). In order to complying with animal welfare guidelines and minimize animal suffering, animals were euthanized with pentobarbitone sodium before tissue sampling.

### Biological samples

After observation of clinical symptoms and anatomical examination, three normal Hubbard broilers (♂) and three VVD Hubbard broilers (♂) that were 35 days old were selected from Zhonghong Sanrong Group Co., Ltd.; Tangshan, China. The birds were euthanized with pentobarbitone sodium. The spleen was collected from the normal birds and the VVD birds, immediately frozen in liquid nitrogen, and stored at –80 ˚C until the RNA was extracted.

## Total RNA isolation and construction of RNA-seq libraries

Six RNA libraries were constructed by extracting the total RNA from the spleens of 3 normal birds (JS) and 3 VVD birds (BS). The NanoPhotometer® spectrophotometer and RNA Nano6000 Assay Kit from the Agilent Bionalyzer 2100 system were used to detect the concentration and integrity of total RNA. Qualified RNA samples were then used for library preparation. For the construction of the RNA-seq libraries, approximately 3 μg total RNA per sample was used. Ribosomal RNA (rRNA) was first removed from the total RNA using the Epicentre Ribo-zero™ rRNA Removal Kit (Epicentre, USA). Free residue of rRNA was cleaned up using ethanol precipitation. The sequencing libraries were then generated from depleted RNA using the reagent kit for Illumina's nebnext® ultra™ directed RNA library (NEB, USA). Then, using random hexamer primer, M-MuLV Reverse Transcriptase, DNA Polymerase I and RNase H, a strand of cDNA was synthesized. The library fragments were purified with AMPure XP system (Beckman Coulter, Beverly, USA) to select cDNA fragments of preferentially 150~200 bp in length. Finally, the products were purified (AMPure XP system) after PCR, and the quality of the library was assessed using the Agilent Bioanalyzer 2100 system. The libraries meeting the quality criteria were sequenced using the Illumina Hiseq 4000 platform from Novegene Bioinformatics Technology Co., Ltd. (Beijing, China), which generated paired-end reads of 150bp. In order to better understand and explore the relevant research, the RNA-Seq datasets supporting the conclusions of this article are available at NCBI project PRJNA644292 with accession number SRR12153415, SRR12153416, SRR12153417, SRR12153418, SRR12153419, SRR12153420.

## Sequencing data analysis and coding potential analysis

Clean reads were obtained by removing poly-N and low-quality reads from the raw data. Q20, Q30 and GC content were subsequently calculated based on the clean reads. The Gallus gallus reference genome was downloaded directly from the genome website (http://asia.ensembl.org/index.html) and HISAT2 was used to align the chicken reference genome with the clean reads. Based on the reference sequence, mapped reads for each sample were assembled by StringTie (V1.3.1) [19]. Four tools were used to distinguish mRNA from lncRNA, namely the Coding Potential Calculator (CPC) [20, 21], Pfam Scan (PFAM) [21, 22], phylogenetic codon substitution frequency (PhyloCSF) [23], and Coding Noncoding Index (CNCI) [24]. Based on these methods, transcripts with coding potential predicted by any one of four tools were filtered out, while transcripts without coding potential were designated as novel lncRNA. The conservation scores of lncRNAs and mRNAs were assessed by Phast (v1.3).

## Target genes (*cis* and *trans*) prediction and differential expression analysis

The target genes of differentially expressed lncRNAs were predicted by *cis* and *trans* analysis. Coding genes 100kb upstream and downstream of lncRNA were considered to be potential *cis* targets because lncRNA can act on neighboring target genes. This is the role of *cis* target gene prediction. Potential *trans* targets were identified by expression levels and Pearson's correlation coefficient ($|r| > 0.95$). The FPKMs (fragments per kilo-base of exon per million fragments mapped) of lncRNAs and mRNAs in each sample were calculated by Cuffdiff (v2.1.1) [25]. The BS/JS group of gene expression level was used to compare with Ballgown [26]. Two criteria must be met for differentially expressed genes and lncRNAs: $|log2FoldChange| \geq 1$ and adjusted $P$ value $< 0.05$.

## Functional enrichment analysis and lncRNA-mRNA network construction

The differentially expressed genes or lncRNAs target genes were used for Gene Ontology (GO) enrichment by the GOseq R package (Release2.12) [27]. The Kyoto Encyclopedia of Gene and Genomes database (KEGG, http://www.genome.jp/kegg) is a database resource for understanding high-level functions and utilities of biological systems [28]. This database, as well as analysis with the software KOBAS (v2.0) [29], was used to identify pathways related to the target genes. Significantly enriched GO terms and KEGG pathways were calibrated for corrected *P*-value < 0.05.

Interactive analysis of lncRNA target genes was predicted by *cis* and *trans* with differentially expressed mRNAs to find the potential relationship between them. LncRNA-mRNA networks were mapped by Cytoscape 3.4.0.

## Quantitative PCR analysis

The expression levels of 16 DE genes were quantified by q-PCR to validate the reproducibility of high-throughput sequencing [30]. The reaction system volume was 10 ul containing 5 ul of SYBR® Premix Ex TaqII (TaKaRa, Dalian, China), 3 ul of RNase-free water, 1 ul cDNA and 0.5 ul of each primer. qPCR was performed on a LightCycler® 96 real time fluorescent quantitative PCR instrument (Roche Applied Science, Indianapolis, USA) and the cycling conditions were as follows: 95 ˚C for 3 min (initial denaturation); followed by 30 cycles at 95 ˚C for 15 s, 60 ˚C for 30 s, 72 ˚C for 30 s; and finally 10 min extension at 72 ˚C. All reactions were performed in triplicate. The relative quantifications of genes were analyzed by the $2^{-\Delta\Delta Ct}$ method. NCBI Primer-Blast was used to design the primers with chicken GAPDH as an endogenous control [31] (S1 Table).

# Results

## Overview of RNA-sequencing data quality

In all, 78.45 gigabytes (GB) of RNA-Seq data sets remained after eliminating the low-quality reads, from which the read yield ranged from 11.78 GB to 15.5 GB. The percentages of Q20 and Q30 were 96.62% and 91.63%, respectively. GC content ranged from 45.72% to 48.44%. Clean reads were aligned with the chicken reference genome and any reads not properly mapped were removed, resulting in uniquely mapped rates of 85.03 to 88.2% (S2 Table).

## Identification and genomic feature analysis of lncRNA and mRNA

In order to identify lncRNAs with high confidence, putative lncRNAs were screened with a series of bioinformatics filter lines (S1 Fig). Four bioinformatic tools were used to assess the coding potential of transcripts. Transcripts of uncertain coding potential were classified as TUCPs and noncoding transcripts were identified as novel lncRNAs. After screening 30252 mRNAs, 3171 TUCPs and 7943 novel lncRNAs (Fig 1A, S2 Fig and S3 Table) were obtained. The novel lncRNAs were classified as 3519 lncRNAs (44.3%), 1056 anti-sense lncRNAs (13.3%) and 3368 intronic lncRNAs (42.4%) (Fig 1B). These transcripts were used for subsequent analysis. To study the genomic characteristics of these transcripts, the transcript length, exon number, open reading frame (ORF) length, and sequence conservation of lncRNAs and mRNAs were compared. In this study, the predicted lncRNAs were shorter in length and ORF, had fewer exons and lower conservation than the mRNAs (Fig 2A–2C). This corresponds with the other lncRNAs.

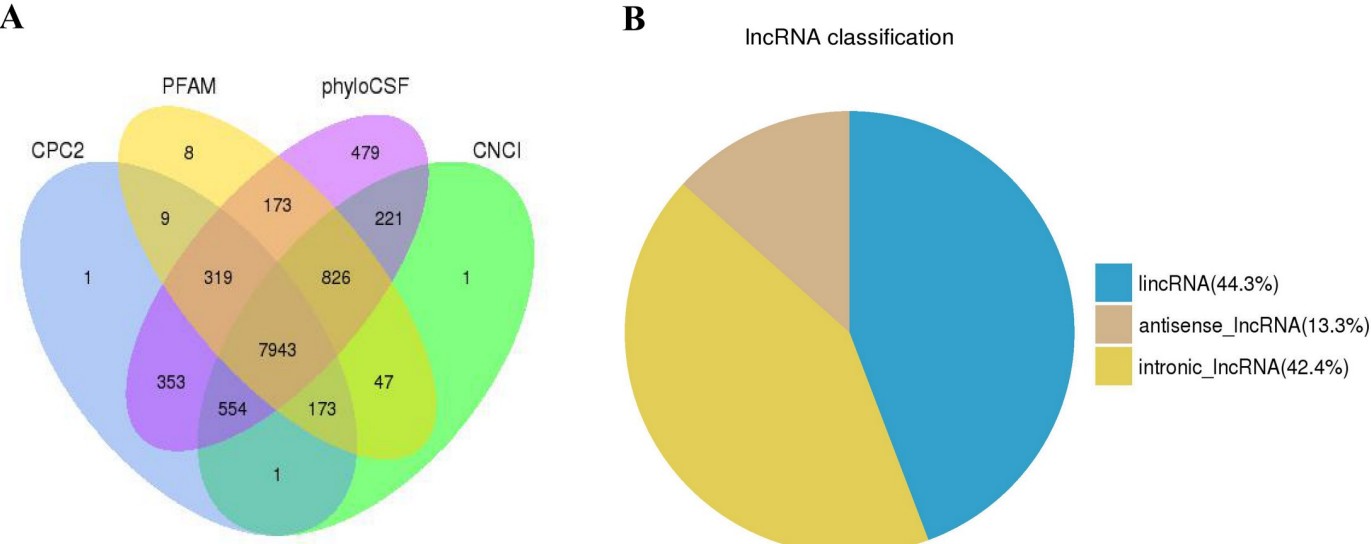

**Fig 1.** (A) Identification of 7943 novel lncRNAs without protein-coding potential identified by CPC, PFAM, phyloCSF, and CNCI. (B) Classification of lncRNAs.

## Expression patterns analysis of lncRNAs and mRNAs

To identify the expression levels of lncRNAs and mRNAs, edgeR was used to screen DE lncRNAs and mRNAs by FPKM. The results showed that the expression levels of lncRNAs were lower than the mRNAs (Fig 2D), which is in agreement with other species. The screening threshold for different types of transcripts was set as $q$-value$< 0.05$ and $|log2FC| \geq 1$. First, DE lncRNAs and protein coding genes in normal birds and VVD birds were focused by us. A total of 124 DE lncRNAs were identified from JS and BS, of which 82 were upregulated and 42 were downregulated; 38 DE lncRNAs were uniquely expressed in BS (Fig 3A and S4-1 Table). A total of 420 DE mRNAs were found; 232 were upregulated in BS, while 188 were downregulated, and 84 of DE genes were only expressed in BS (Fig 3B and S4-2 Table). To explore the location of these DE lncRNAs and DE mRNAs, the differential expression patterns of lncRNAs and mRNAs were shown by circular chromosome distribution (Figs 4 and 5). These DE mRNAs were distributed among nearly all chromosomes. DE lncRNAs were mainly distributed in chromosomes 1 and 2. In addition, systematic cluster analysis of differential expression lncRNAs and mRNAs between BS/JS groups were revealed by heat map. Hierarchical clustering analysis showed that the expression pattern of lncRNAs (Fig 6A) and mRNAs (Fig 6B) were similar.

## Functional enrichment analysis of lncRNAs

GO and KEGG analysis of the role of lncRNAs in *cis* and *trans* regulation was used to further understand the potential function of DE lncRNAs [32, 33]. The significantly enriched GO terms by *cis* target genes were shown in (Fig 7A). GO enrichment analysis indicated that the *cis* target genes by DE lncRNAs were significantly enriched in 61 GO terms (corrected *P*-value < 0.05, S5-1 Table). These GO terms were mainly related to interferon receptors, phosphorylation and immune regulation such as interferon-alpha/beta receptor binding, positive regulation of peptidyl-serine phosphorylation of STAT protein, natural killer cell activation involved in immune response, serine phosphorylation of STAT protein, humoral immune response, and so on. The 20 most significantly enriched KEGG pathways were shown in

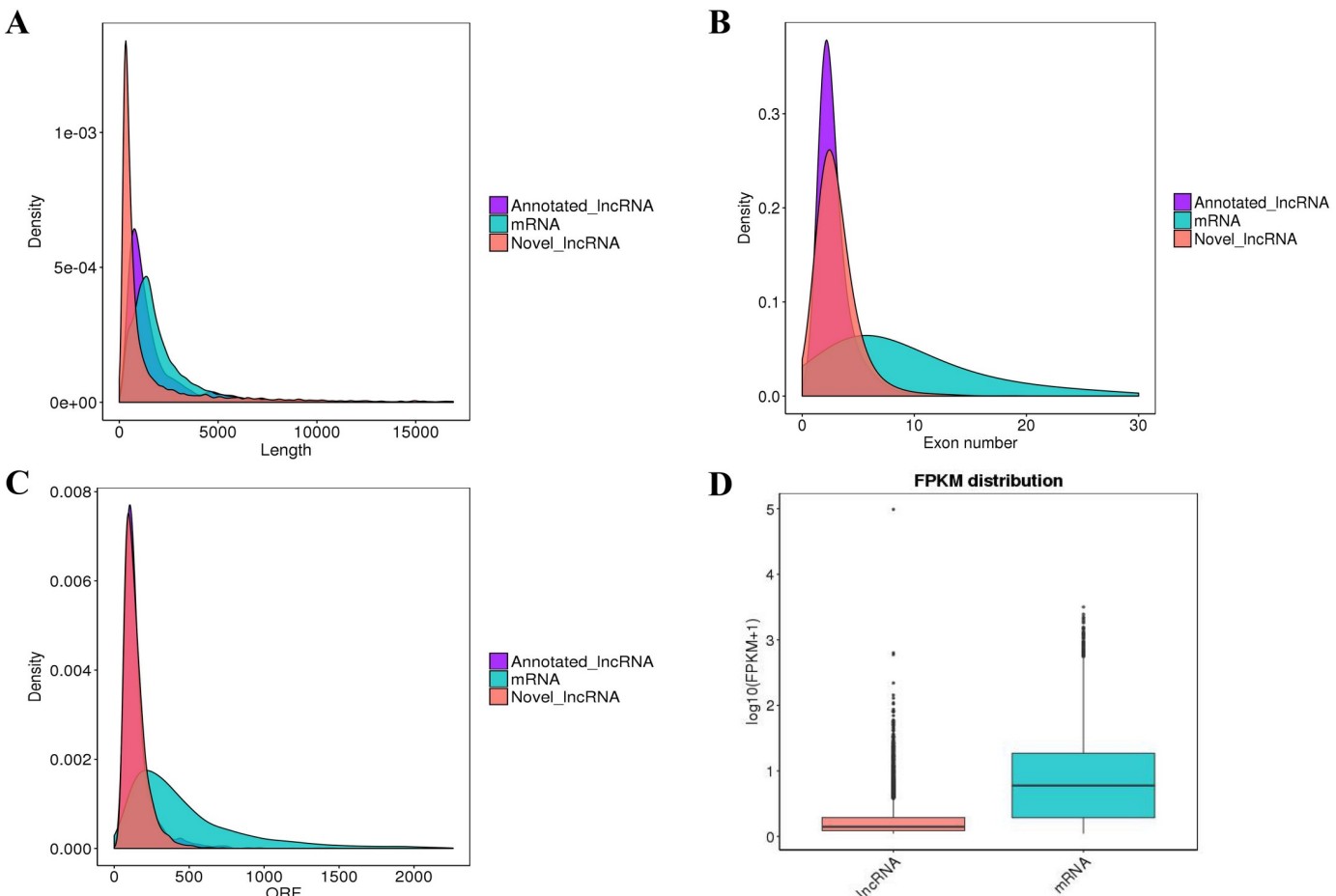

**Fig 2. Genomic characteristic of lncRNAs and mRNAs.** (A) Length distribution of lncRNAs and mRNAs; (B) Exon number distribution of lncRNAs and mRNAs; (C) ORF length distribution of lncRNAs and mRNAs; (D) The expression levels of lncRNAs and mRNAs in BS and JS groups.

Fig 7B. We focused on the following pathways with the highest enrichment of target genes: the regulation of autophagy, RIG-I-like receptor signaling pathway, Jak-STAT signaling pathway, Toll-like receptor signaling pathway, Cytokine-cytokine receptor interaction, TGF-beta signaling pathway and regulation of actin cytoskeleton. These pathways might be significantly associated with VVD (S6-1 Table).

According to Pearson's correlation coefficients ($|r| > 0.95$), we predicted the potential target genes of lncRNAs in *trans* regulation. These *trans* target genes of lncRNAs with significantly enriched GO terms are shown in (Fig 8A and S5-2 Table). These GO terms were mainly related to cell regulation and are similar to lncRNAs in the *cis* role. The 20 most enriched KEGG pathways for *trans* target genes were shown in Fig 8B. We focused on the following pathways for *trans* targets of lncRNAs: the adherens junction, focal adhesion, oxidative phosphorylation, ECM-receptor interaction, regulation of actin cytoskeleton, insulin signaling pathway, TGF-beta signaling pathway and mTOR signaling pathway (S6-2 Table).

## Functional enrichment analysis of mRNAs

Through comparison of the BS vs. JS groups, we obtained 420 differentially expressed mRNAs. GO enrichment analysis was performed on 420 DE mRNAs. There were 356 genes

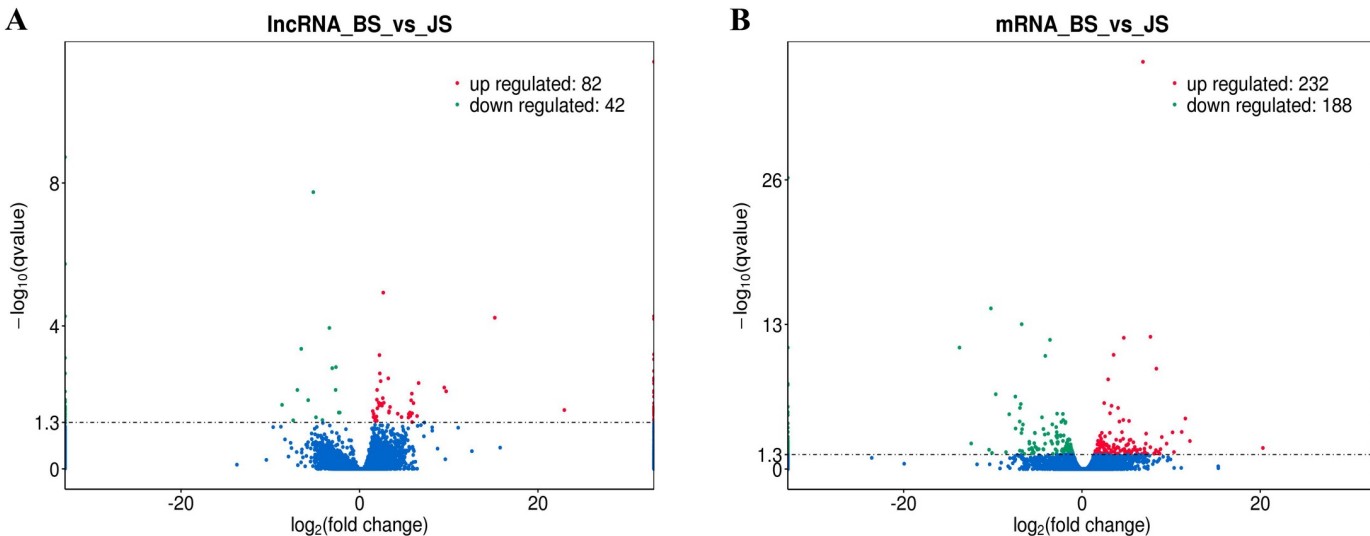

**Fig 3.** (A) Volcano plot shown DE-lncRNAs in VVD and normal birds. (B) Volcano plot showing DE-mRNAs in VVD and normal birds. Green and red represent downregulated and upregulated expression, respectively.

corresponding to DE mRNAs with GO annotation. Using GO analysis these genes were classified into broad categories of biological process (BP), molecular function (MF) and cellular component (CP). The top 20 enriched terms and KEGG pathways are shown in Fig 9. Enrichment analysis revealed that a total of four highly enriched GO terms were obtained from BS vs. JS (corrected *P*-value < 0.05, S5-3 Table). Through KEGG analysis, we focused on the following signaling pathways related to skeletal development and immunity: the Jak-stat signaling pathway, Toll-like receptor signaling pathway, Wnt-signaling pathway, mTOR signaling pathway, VEGF signaling pathway, Notch signaling pathway, TGF-beta signaling pathway and Fanconi anemia pathway (S6-3 Table).

## Association analysis between lncRNA and mRNA

To identify key molecular players in the development of VVD, we first constructed a co-expression network of DEmRNAs and lncRNAs. All 11 DE genes (*ERGIC1*, *GARS*, *PLXNB1*, *GREB1L*, *REV1*, *ENSGALG00000017040*, *ENSGALG00000030736*, *ENSGALG00000034144*, *TM4SF1*, *NFIC*, *PDIA4*) were obtained from the analysis (Fig 10A and S7 Table). The co-expression network is shown in Fig 10B. Furthermore, we screened out genes enriched in the Jak-stat signaling pathway, Toll-like receptor signaling pathway, Wnt-signaling pathway, mTOR signaling pathway, VEGF signaling pathway, Notch signaling pathway, TGF-beta signaling pathway and Fanconi anemia pathway, since these pathways are related to immunity and skeleton development. A total of 30 candidate DE genes were obtained from these pathways (S8 Table). We then analyzed the interaction between the DE genes and their corresponding lncRNAs; the co-expression network is shown in Fig 11.

## qPCR validation

To validate the RNA-seq result, the expression levels of 16 dysregulated genes were confirmed by qPCR, including eight lncRNAs (*XLOC_068551*, *ALDBGALG0000000649*, *XLOC_037307*, *XLOC_015443*, *XLOC_031206*, *XLOC_043086*, *XLOC_089730*, *XLOC_064330*) and eight mRNAs (*MACF1*, *MYO1B*, *NNT*, *FKBP5*, *TNS1*, *JCHAIN*, *IFI6*, *SEC61G*). The total cDNA of

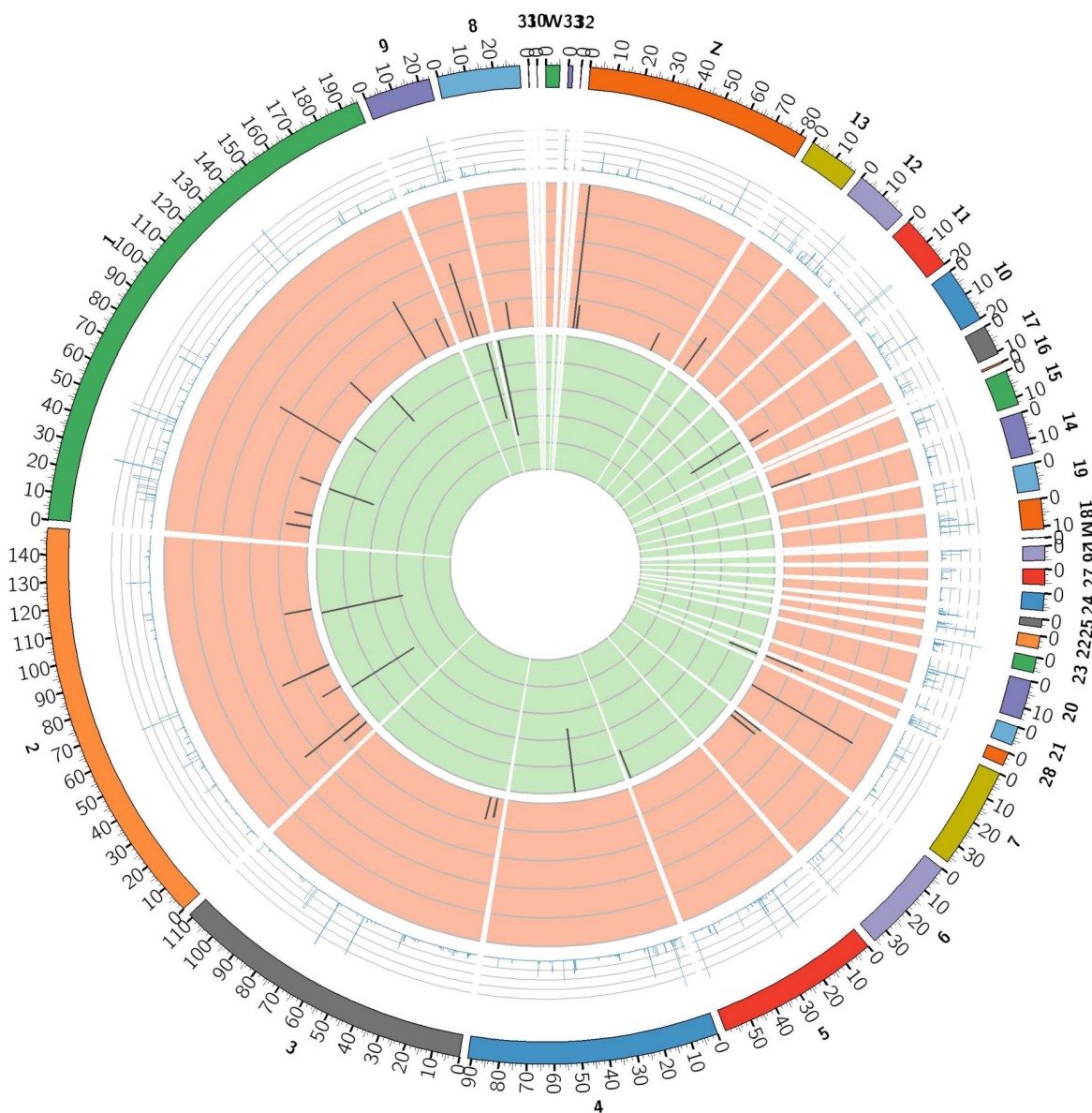

**Fig 4. Circular chromosome distribution of DE-lncRNAs.** The outermost circle displays chicken chromosomes. The second circle displays average FPKM. The remaining circles display the significantly upregulated and downregulated transcripts.

the six individuals in the BS/JS group was used for quantitative analysis. The results of qPCR were identical with the RNA-seq (Fig 12), showing that the RNA-seq has high reliability and accuracy.

## Discussion

Broiler leg disease has been frequently described in recent years. Its pathogenesis is complex. However, there is a paucity of studies on the molecular mechanisms of VVD, and the etiology of VVD in broilers has not been fully elucidated [34, 35]. Therefore, it is important to determine the molecular mechanism of VVD. The heritable nature and bone quality traits of VVD have been reported. Heritability of leg problems ranged from 0.10 to 0.13 and that of bone

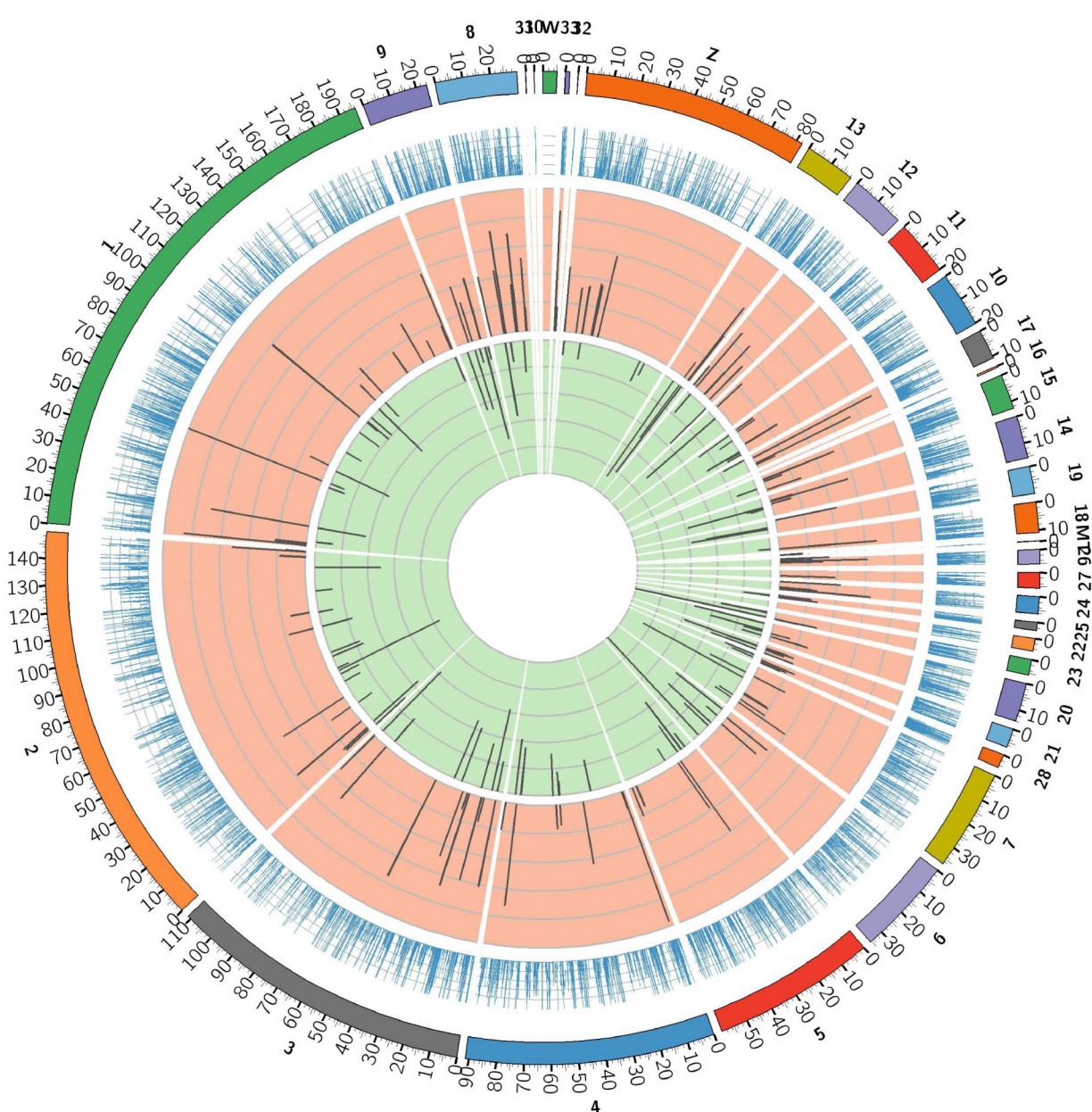

**Fig 5. Circular chromosome distribution of DE-mRNAs.** The outermost circle displays chicken chromosomes. The second circle displays average FPKM. The remaining circles display the significantly upregulated and downregulated transcripts.

quality ranged from 0.10 to 0.77 [3, 36–38]. And the prevalence of VVD has declined through genetic selection [1, 6, 39]. In our research, we performed transcriptome sequencing of six chicken spleen samples from BS and JS groups. The main purpose was to identify the lncRNAs and mRNAs that are related to VVD, and their potential functions. A total of 1951 annotated lncRNAs, 7943 novel transcripts, 3171 TUCPs, and 30252 mRNAs were identified in the spleen samples. We obtained 420 DE mRNAs, of which 232 were upregulated and 188 were downregulated in VVD birds. 124 DE lncRNAs were also found, of which 82 were upregulated and 32 downregulated. These data provide a valuable resource for the analysis of molecular mechanisms of VVD and posttranscriptional regulation.

**A**                                                                                             **B**

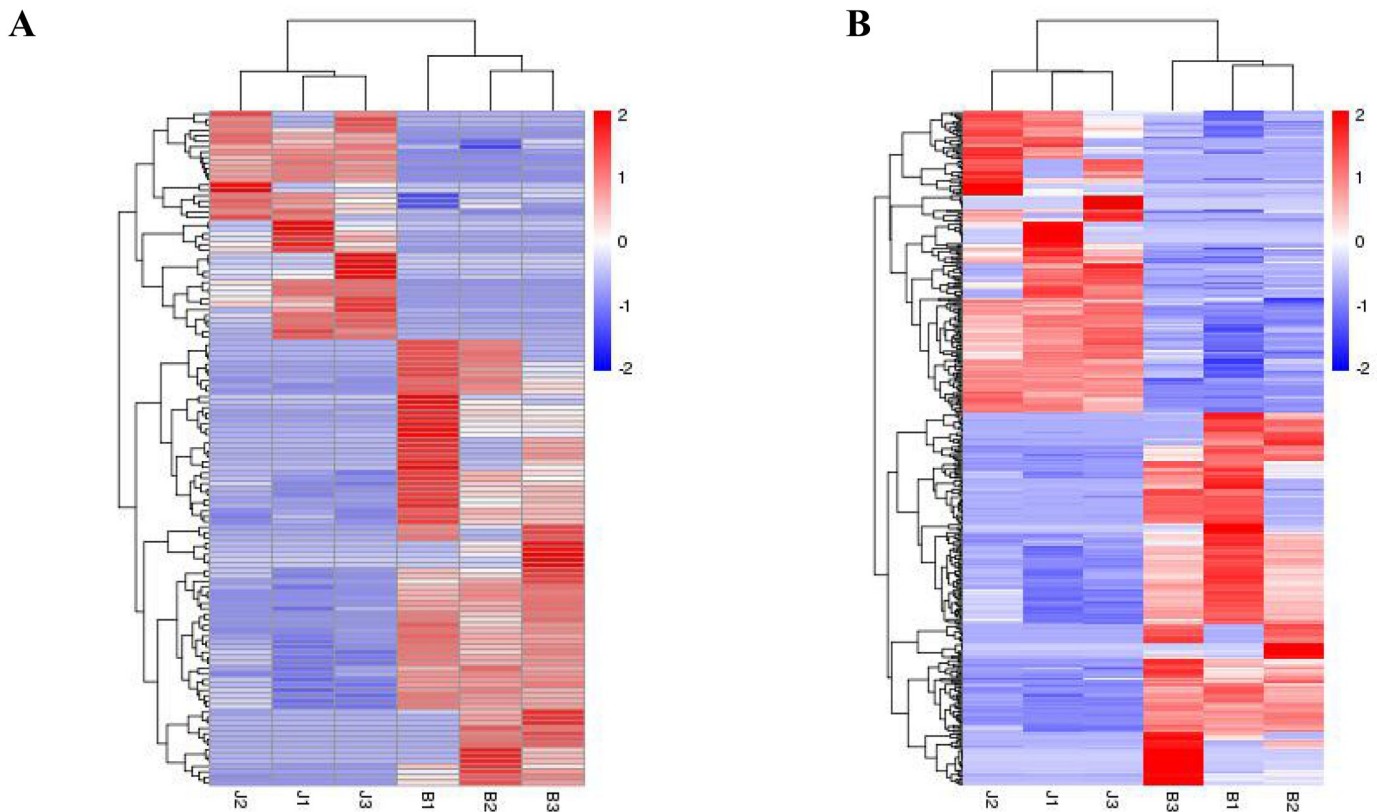

**Fig 6. Cluster analyses of DE-lncRNAs (A) and mRNAs (B).** Red means a relatively higher level of expression; Blue means a relatively lower level of expression.

The Hubbard broiler was developed by the Hubbard company of the United States. The Hubbard broiler grows fast and has high meat yield, but the incidence of leg disease is also high. In previous studies, leg disorders were associated with compromised growth, lipid metabolism, bone quality and structure [7]. At present, the molecular mechanism of VVD is poorly understood. Differentially expressed genes and lncRNAs may play important regulatory roles in VVD. In our study, through analysis of the co-expression network of lncRNAs and mRNA, a total of 11 DE genes were identified. Remarkably, among the interaction networks, *GARS* is an important target and was predicted by 32 lncRNAs. This gene is classically linked to predominant distal hereditary motor neuropathies (dHMN). It is related to motor neuron disorders (MND-amyotrophic lateral sclerosis) in humans. The genetic factors of this disease are accepted worldwide [40]. In addition, studies have found a missense mutation in *GARS*, which is considered to be the cause of Charcot Marie Tooth (CMT) disease, which is characterized by sensory loss, areflexia, distal muscle weakness and atrophy [41]. Notably, *NFIC* also is an important target and was predicted by 28 lncRNAs. NFI proteins are related to cell growth and disease state of *NFIC*, which is a member of the NFI family. Studies have shown that *NFIC* is closely linked to osteogenic differentiation, and found that the expression of *NFIC* in osteoblasts of patients with human osteoporosis was decreased [42, 43], but this is still in the stage of basic research and there is little research on poultry. These findings suggest, however, that these genes may play a key role in VVD, thus providing a new entry point for further understanding of the regulation mechanisms of VVD.

To gain insight into the interactions of genes, lncRNAs and pathways associated with VVD, we screened 30 DE genes in eight pathways (the Jak-stat signaling pathway, Toll-like receptor

A

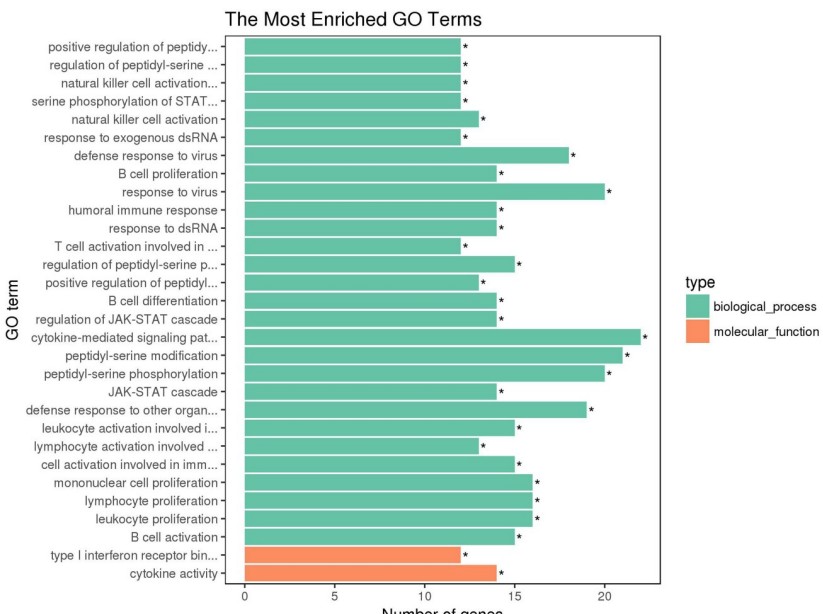

B

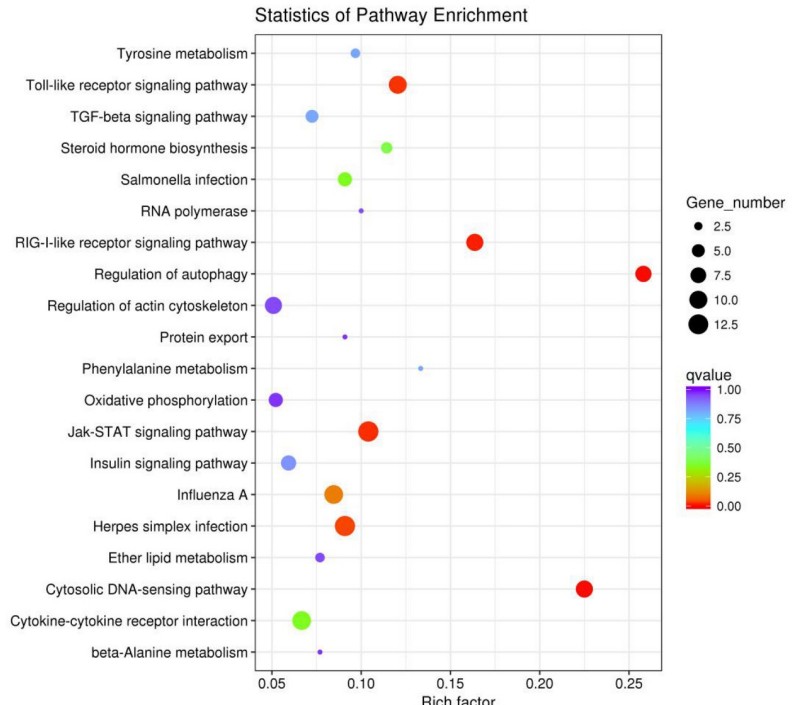

**Fig 7. The functional enrichment analysis of *cis* target genes of the DE-lncRNAs.** (A) Gene ontology (GO) function annotation of *cis* target genes of the DE-lncRNAs. The Y-axis means the detailed terms and the X-axis means the gene numbers. (B) The top 20 Kyoto Encyclopedia of Genes and Genomes (KEGG) pathways of *cis* target genes of the DE-lncRNAs. The Y-axis indicates the name of KEGG pathway and the X-axis indicates the gene ratio. The size of the dots represents the numbers of target genes, and the color of the dots represents *P*-value.

A

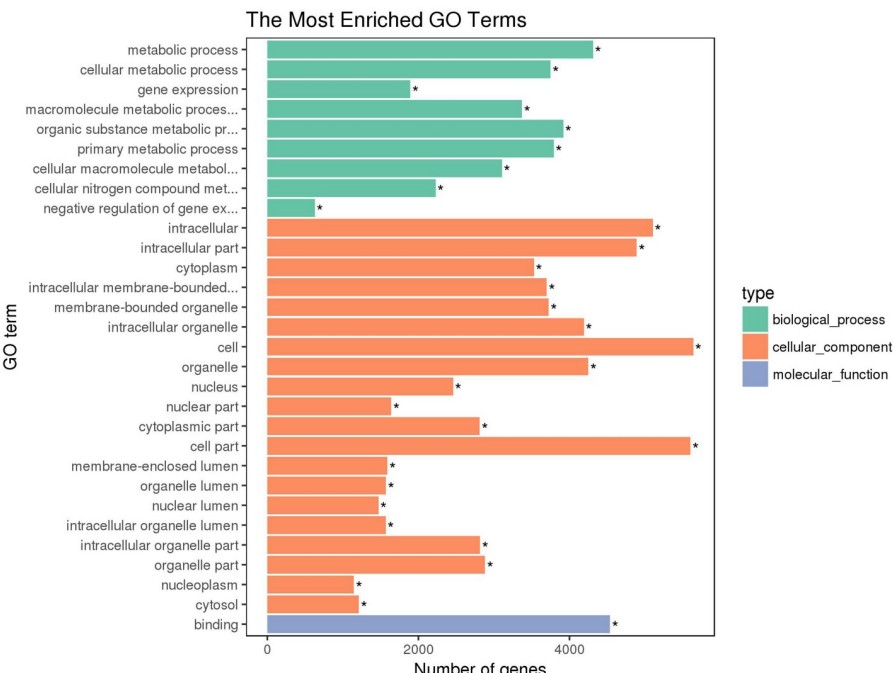

B

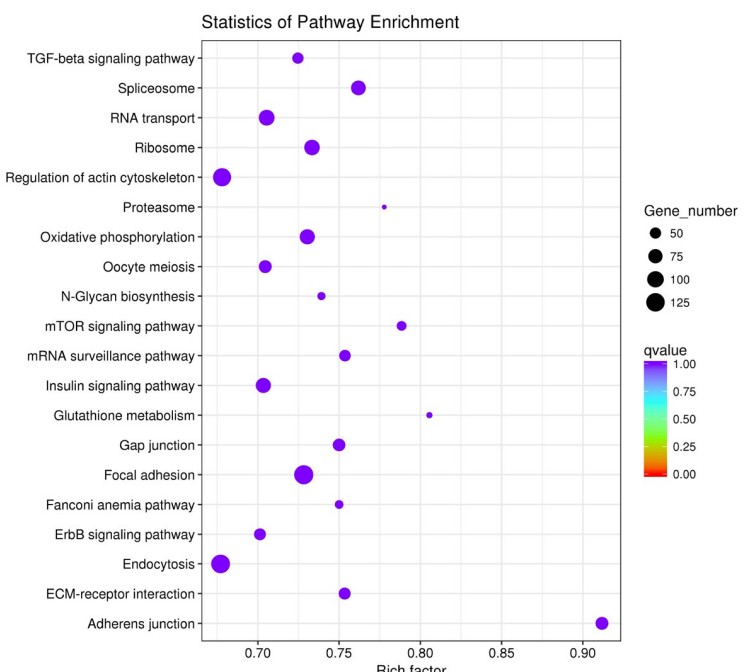

**Fig 8. The functional enrichment analysis of *trans* target genes of the DE-lncRNAs.** (A) Gene ontology (GO) function annotation of *trans* target genes of the DE-lncRNAs. The Y-axis means the detailed terms and the X-axis means the gene numbers. (B) The top 20 Kyoto Encyclopedia of Genes and Genomes (KEGG) pathways of *trans* target genes of the DE-lncRNAs. The Y-axis indicates the name of KEGG pathway and the X-axis indicates the gene ratio. The size of the dots represents the numbers of target genes, and the color of the dots represents *P*-value.

A

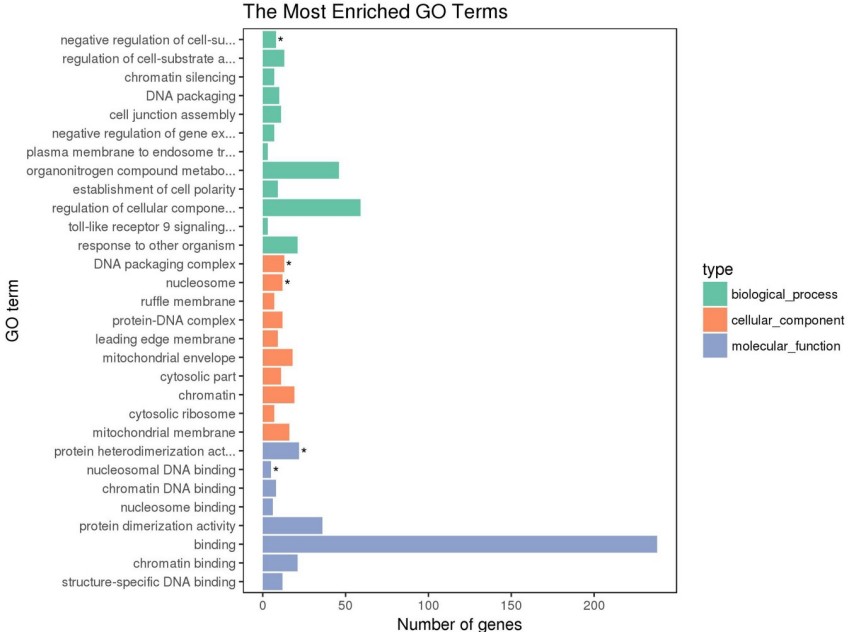

B

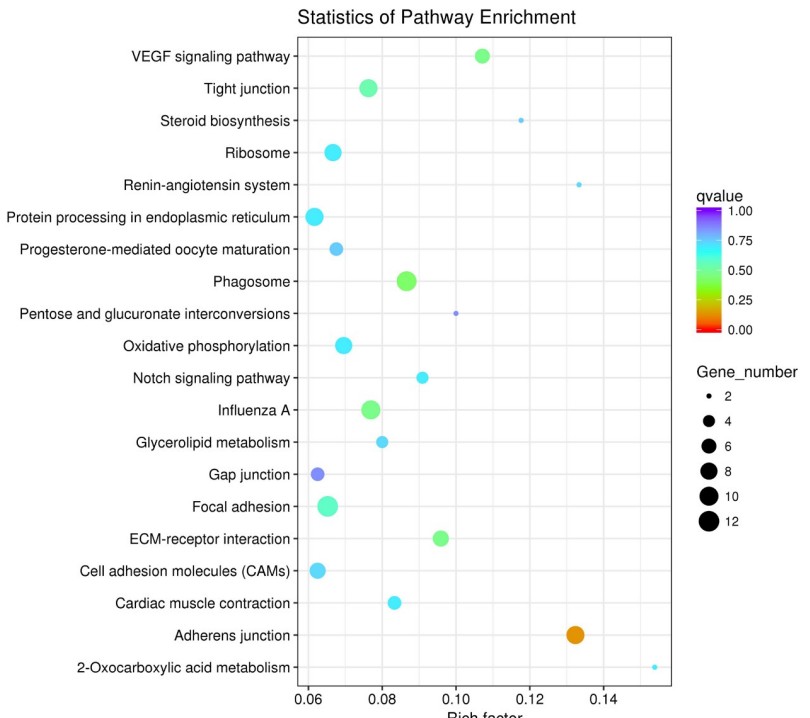

**Fig 9. The functional enrichment analysis of the DE-mRNAs.** (A) Gene ontology (GO) function annotation of the DE-mRNAs. The Y-axis indicates the detailed terms and the X-axis indicates the gene numbers. (B) The top 20 Kyoto Encyclopedia of Genes and Genomes (KEGG) pathways of the DE-mRNAs. The Y-axis indicates the name of KEGG pathway and the X-axis indicates the gene ratio. The size of the dots represents the numbers of genes, and the color of the dots represents *P*-value.

A

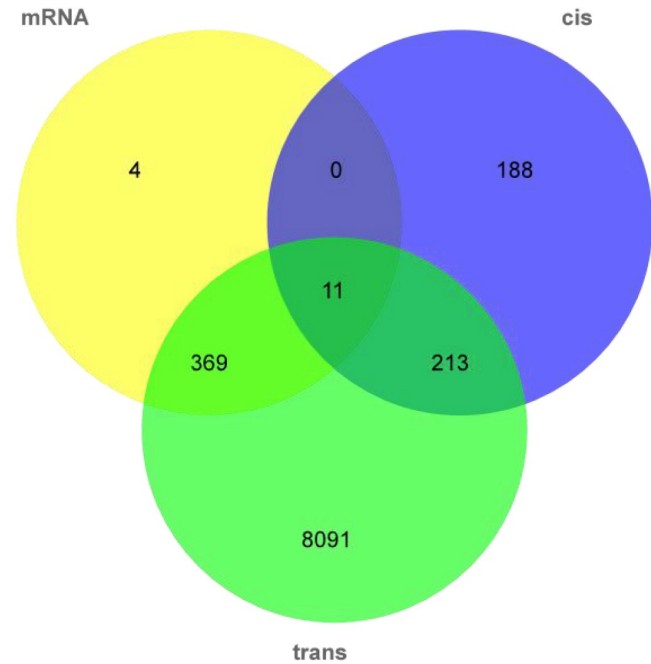

B

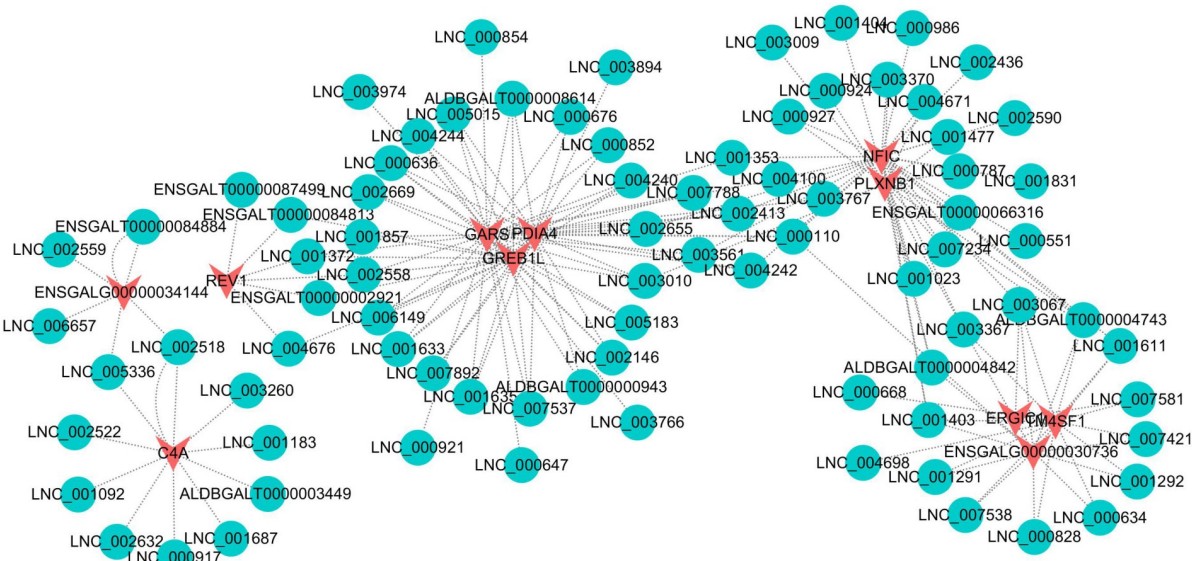

**Fig 10. Analysis of DE mRNA-lncRNA interaction network.** (A) Venn diagram indicates relationships between lncRNAs and mRNAs. (B) DE mRNA-lncRNA regulatory networks. Red nodes represent mRNAs; blue nodes represent lncRNAs.

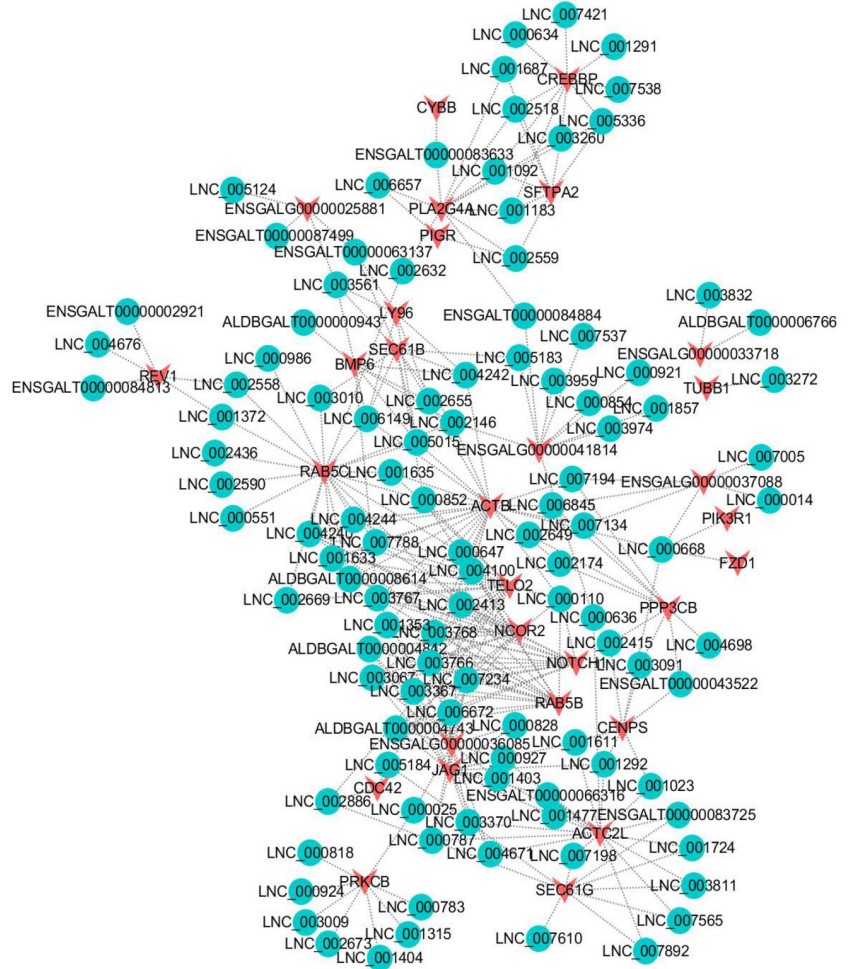

**Fig 11. Interaction network analysis of DE mRNA-lncRNA that is enriched in bone or immune pathways.** Red nodes represent mRNAs; blue nodes represent lncRNAs.

signaling pathway, Wnt-signaling pathway, mTOR signaling pathway, VEGF signaling pathway, Notch signaling pathway, TGF-beta signaling pathway, and Fanconi anemia pathway), which are related to immunity and skeleton development. The interaction between the DE genes and the lncRNAs was analyzed. Of the node genes, *NOTCH1* attracted attention, as it was predicted by 11 lncRNAs. The Notch signaling pathway was conserved. The Notch pathway, which regulates biological development through interaction between cells, plays an important role in cell proliferation, differentiation, development and homeostasis [44]. The Notch pathway can interact with many growth factors and cytokines, such as VEGF [45], Wnt/β-catenin [46], TGF-β [47, 48] and so on. Nobta's study found that *NOTCH1* could promote the osteogenic activity of osteoblasts [49]. ACTB as a top2 DE gene is also an important target, which was predicted by 26 lncRNAs. *ACTB* encodes β-actin, an abundant cytoskeletal housekeeping protein which can result in a pleiotropic developmental disorder after loss-of-function mutations. Studies have found that missense mutations of *ACTB* may cause Baraitser-Winter syndrome (BRWS) in humans, characterized by intellectual disability, cortical malformations, coloboma, organ malformation [50, 51]. Bone morphogenetic protein 6 (*BMP6*) is another key gene that we focused on. Bone morphogenetic proteins (BMPs) are growth factors

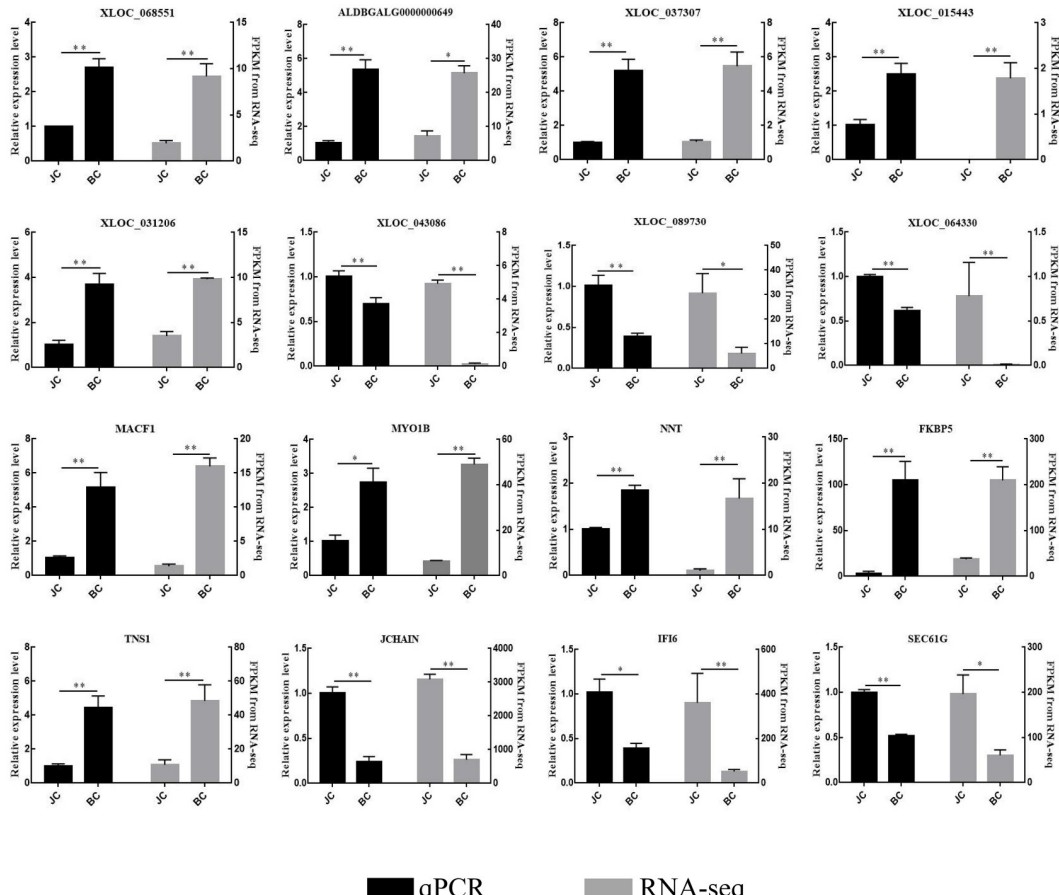

**Fig 12. Validation of RNA-seq results by RT-qPCR with GAPDH as the endogenous control.** Expression level was calculated using the comparative cycle threshold ($2^{-\Delta\Delta Ct}$) value method. The data were expressed as the mean±SEM ($*P < 0.05$; $**P < 0.01$).

and play an important role during skeletal development [52]. *BMP6*, as a member of the BMP family, can effectively induce the formation of bone. Notably, *BMP6* has a certain effect on the immunoregulatory function of bone marrow mesenchymal stem cells (BMMSCs) [53]. These results indicate that the key core node genes are important connection points between pathways, and this interaction network of pathways provides a new understanding of the molecular regulation mechanism underlying VVD in broilers.

In summary, we identified differentially expressed mRNAs and lncRNAs from the spleen of Hubbard broilers in BS/JS groups, and we constructed the interaction network among mRNAs, lncRNAs and signaling pathways involved in immunity and skeleton development. These results provide a basis for further study of the underlying molecular mechanism of VVD. Further studies should verify the function of these key genes and interaction networks at the cellular level.

## Conclusion

This study used RNA-Seq technology to analyze the spleen transcriptome in VVD Hubbard broilers and normal Hubbard broilers. The results revealed lncRNA-mRNA networks, novel genes and pathway enrichment that involved in immunity and bone development about

Hubbard broilers. These findings will provide some reference for the etiology and pathogenesis of VVD.

## Supporting information

**S1 Fig. Identification pipeline for lncRNAs, TUPCs and mRNAs.**
(TIF)

**S2 Fig. 7943 without protein-coding potential lncRNAs were selected by Cufflflinks and Scripture.**
(TIF)

**S1 Table. Primers of lncRNAs and mRNAs for qPCR validation.**
(XLSX)

**S2 Table. Summary of RNA-seq data and reads mapped to the *Gallus gallus* reference genome.**
(XLSX)

**S3 Table. Characteristics of mRNAs, TUCPs and lncRNAs.**
(XLSX)

**S4 Table. List of differentially expressed lncRNAs and mRNAs.**
(XLSX)

**S5 Table. GO enrichment analysis of the significantly differences lncRNAs target genes and mRNAs between BS and JS groups.**
(XLSX)

**S6 Table. KEGG enrichment analysis of the significantly differences lncRNAs target genes and mRNAs between BS and JS groups.**
(XLSX)

**S7 Table. Focused on KEGG pathways of the target genes, *cis* and *trans*, of differentially expressed lncRNAs.**
(XLSX)

**S8 Table. Focused on KEGG pathways of differentially expressed mRNA.**
(XLSX)

## Author Contributions

**Data curation:** Hehe Tang.

**Formal analysis:** Hehe Tang.

**Supervision:** Yaping Guo, Zhenzhen Zhang, Zhuanjian Li, Yanhua Zhang, Yuanfang Li, Xiangtao Kang.

**Validation:** Hehe Tang.

**Writing – original draft:** Hehe Tang, Ruili Han.

**Writing – review & editing:** Hehe Tang.

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
