## [Decision Letter · Decision Letter 0]

1 Jul 2020

PONE-D-20-04909

Integrative Analysis of Long Non-coding RNA and mRNA in Broilers with Valgus-varus Deformity

PLOS ONE

Dear Dr. han,

Thank you for submitting your manuscript to PLOS ONE. After careful consideration, we feel that it has merit but does not fully meet PLOS ONE’s publication criteria as it currently stands. Therefore, we invite you to submit a revised version of the manuscript that addresses the points raised during the review process. Please make sure to address Reviewer #2 concerns

We look forward to receiving your revised manuscript.

Kind regards,

Carlos M. Isales, M.D.

Academic Editor

PLOS ONE

Journal Requirements:

2. In your Methods section, please provide additional information on the animal sacrifice and ensure you have included details on : (1) whether the method of broiler sacrifice( by jugular vein severance) is a standard (2) whether this method is approved by your ethics committee (3) what methods/efforts are used to alleviate animal suffering and ensure the death is completed.

3. We note that you are reporting an analysis of a microarray, next-generation sequencing, or deep sequencing data set. PLOS requires that authors comply with field-specific standards for preparation, recording, and deposition of data in repositories appropriate to their field. Please upload these data to a stable, public repository (such as ArrayExpress, Gene Expression Omnibus (GEO), DNA Data Bank of Japan (DDBJ), NCBI GenBank, NCBI Sequence Read Archive, or EMBL Nucleotide Sequence Database (ENA)). In your revised cover letter, please provide the relevant accession numbers that may be used to access these data. For a full list of recommended repositories, see http://journals.plos.org/plosone/s/data-availability#loc-omics or http://journals.plos.org/plosone/s/data-availability#loc-sequencing.

Additional Editor Comments (if provided):

Reviewers' comments:

Reviewer's Responses to Questions

**Comments to the Author**

1. Is the manuscript technically sound, and do the data support the conclusions?

Reviewer #1: Yes

Reviewer #2: Yes

Reviewer #3: Yes

Reviewer #4: Yes

2. Has the statistical analysis been performed appropriately and rigorously? 

Reviewer #1: Yes

Reviewer #2: Yes

Reviewer #3: Yes

Reviewer #4: Yes

3. Have the authors made all data underlying the findings in their manuscript fully available?

Reviewer #1: Yes

Reviewer #2: Yes

Reviewer #3: Yes

Reviewer #4: No

4. Is the manuscript presented in an intelligible fashion and written in standard English?

Reviewer #1: Yes

Reviewer #2: Yes

Reviewer #3: Yes

Reviewer #4: Yes

5. Review Comments to the Author

Reviewer #1: Methods - Were the birds stunned prior to jugular vein severance? Were any power equations undertaken to assess if 3 birds per group would give representative and accurate data?

No additional comments.

Reviewer #2: Please ensure the all required field of the admission process are followed correctly and completely.

This paper covers an area of global interest, due to the large number of birds intensively reared globally for meat and the lameness issues associated with this type of meat production.

In order to make the paper relevant and engaging to a broad audience, both more detail and more broad, contextual statements are required. Please include information on the number of birds produced annually, the strains of birds must commonly used, and the incidence of specific types of lameness in broilers.

Reference to a country (UK) is not appropriate in a paper that is posing a hypothesis that genetic strain predisposes to VVD: the UK does not have unique broiler genetics: it draws from the same key global genetic resources as other countries. Paper focuses on just one strain of broiler (although the authors do not specify which of the Hubbard strains it is) and draws conclusions about broilers generally: the paper needs to ensure that the introduction and conclusions make it clear that just one strain of broiler has been investigated. The paper would have made a far stronger contribution if the genetic pathways associated with VVD had been studied in a number of commonly used strain of broiler. Nonetheless, this manuscript make a valuable contribution in a field that requires investigation - but its importance is overstated by genericizations to all broilers.

Introduction does not adequately introduce VVD to the reader: current thinking on the multiple contributory causes to VVD is not adequately covered. Leading on from this, given the substantial number of non-genetic factors contributing to VVD, the methodology does not adequately describe the rearing conditions of the birds used: this is essential to demonstrate all environmental factors were controlled and homogenous across all birds, leaving genetic pre-disposition as the remaining factor for assessment.

Bird rearing conditions require the same level of detail as the molecular methodologies. Please include stocking density, lighting regime, environmental controls, vaccinations programme, diet formulation and nutrient specification.

Euthanasia method does not comply with the PLOS1 requirements for adherence to AVMA guidelines for euthanasia of animals:

From the AVMA guidelines:

M3.13.1 Exsanguination Exsanguination can be used to ensure death subsequent to stunning, or in otherwise unconscious animals. Because anxiety is associated with extreme hypovolemia, exsanguination must not be used as a sole means of euthanasia.

Discussion is clear and well written. Just needs some comments to draw the findings back to a wider context, and to address the general point made above.

Reviewer #3: The manuscript describes a useful insight into molecular mechanism and parthenogenesis of VVD. This study can add to genetic selection of broilers to reduce its prevalence. However, the number of samples is insufficient and more samples needed

Reviewer #4: The manuscript provide a novel study for the transcriptome profiles of spleen tissue from 3 normal Hubbard broilers and 3 VVD Hubbard broilers. The findings provide some reference for the etiology and pathogenesis of VVD. There are few areas which require further clarifications and more discussion.

1. The authors should add the relation between VVD and spleen, and declare the significance of spleen transcript sequencing for VVD.

2. Does the sound Hubbard broilers mean the normal broilers? There were two statements in the paper, you should keep it consistent.

3. Since, it is an RNA seq, the authors must provide submission links where the RNAseq raw data is submitted.

4. The format of references need modification, for example, L406, L416, L497……

6. PLOS authors have the option to publish the peer review history of their article (what does this mean?). If published, this will include your full peer review and any attached files.

Reviewer #1: No

Reviewer #2: No

Reviewer #3: No

Reviewer #4: No

---

## [Author Response · Author response to Decision Letter 0]

12 Aug 2020

Thank you very much for working our paper.We have revised on our manuscript according your comments. These comments were valuable and helpful for revising and improving our paper as well as for providing important overall guidance for our study. We have studied the comments carefully and have made corrections, which we hope will meet with your approval.

---

## [Decision Letter · Decision Letter 1]

8 Sep 2020

Integrative Analysis of Long Non-coding RNA and mRNA in Broilers with Valgus-varus Deformity

PONE-D-20-04909R1

Dear Dr. han,

We’re pleased to inform you that your manuscript has been judged scientifically suitable for publication and will be formally accepted for publication once it meets all outstanding technical requirements.

Kind regards,

Carlos M. Isales, M.D.

Academic Editor

PLOS ONE

Additional Editor Comments (optional):

Reviewers' comments:

Reviewer's Responses to Questions

**Comments to the Author**

1. If the authors have adequately addressed your comments raised in a previous round of review and you feel that this manuscript is now acceptable for publication, you may indicate that here to bypass the “Comments to the Author” section, enter your conflict of interest statement in the “Confidential to Editor” section, and submit your "Accept" recommendation.

Reviewer #2: All comments have been addressed

Reviewer #4: All comments have been addressed

2. Is the manuscript technically sound, and do the data support the conclusions?

Reviewer #2: Yes

Reviewer #4: Yes

3. Has the statistical analysis been performed appropriately and rigorously? 

Reviewer #2: Yes

Reviewer #4: Yes

4. Have the authors made all data underlying the findings in their manuscript fully available?

Reviewer #2: Yes

Reviewer #4: Yes

5. Is the manuscript presented in an intelligible fashion and written in standard English?

Reviewer #2: Yes

Reviewer #4: Yes

6. Review Comments to the Author

Reviewer #2: (No Response)

Reviewer #4: The manuscript provide a novel study for the transcriptome profiles of spleen tissue from 3 normal Hubbard broilers and 3 VVD Hubbard broilers. The findings provide some reference for the etiology and pathogenesis of VVD. All the questions have been revised, I agree to recommend publication.

7. PLOS authors have the option to publish the peer review history of their article (what does this mean?). If published, this will include your full peer review and any attached files.

Reviewer #2: No

Reviewer #4: No

---

## [Editor Report · Acceptance letter]

15 Sep 2020

PONE-D-20-04909R1

Integrative Analysis of Long Non-coding RNA and mRNA in Broilers with Valgus-varus Deformity

Dear Dr. Han:

I'm pleased to inform you that your manuscript has been deemed suitable for publication in PLOS ONE. Congratulations! Your manuscript is now with our production department.

Kind regards,

on behalf of

Professor Carlos M. Isales 

Academic Editor

PLOS ONE